# Bone Marrow-Derived VSELs Engraft as Lung Epithelial Progenitor Cells after Bleomycin-Induced Lung Injury

**DOI:** 10.3390/cells10071570

**Published:** 2021-06-22

**Authors:** Andrzej K. Ciechanowicz, Katarzyna Sielatycka, Monika Cymer, Marta Skoda, Malwina Suszyńska, Kamila Bujko, Mariusz Z. Ratajczak, Diane S. Krause, Magdalena Kucia

**Affiliations:** 1Department of Regenerative Medicine, Center for Preclinical Research and Technology, Medical University of Warsaw, 02-097 Warsaw, Poland; andrzej.ciechanowicz@wum.edu.pl (A.K.C.); monika.cymer@wum.edu.pl (M.C.); marta.skoda@ryvu.com (M.S.); mariusz.ratajczak@wum.edu.pl (M.Z.R.); 2Institute of Biology, Faculty of Exact and Natural Sciences, University of Szczecin, 71-415 Szczecin, Poland; katarzyna.sielatycka@usz.edu.pl; 3Stem Cell Institute at James Graham Brown Cancer Center, University of Louisville, Louisville, KY 40202, USA; malwinasuszynska@gmail.com (M.S.); kamila.bujko@wum.edu.pl (K.B.); 4Departments of Laboratory Medicine, Cell Biology and Pathology and the Yale Stem Cell Center, Yale University School of Medicine, New Haven, CT 06509, USA; diane.krause@yale.edu

**Keywords:** alveolar type II cells, bronchioalveolar stem cells, very small embryonic-like stem cells, regenerative medicine, bleomycin-induced injury, lung regeneration

## Abstract

Background: Alveolar type 2 (AT2) cells and bronchioalveolar stem cells (BASC) perform critical regenerative functions in response to lung damage. Published data show that nonhematopoietic, bone marrow-derived “very small embryonic-like stem cells” (VSELs) can differentiate in vivo into surfactant protein C (SPC)-producing AT2 cells in the lung. Here, we test directly whether VSEL-derived BASC and AT2 cells function to produce differentiated progeny. Methods: using a reporter mouse in which the H2B-GFP fusion protein is driven from the murine SPC promoter, we tested whether bone marrow-derived VSELs or non-VSEL/nonhematopoietic stem cells (non-VSEL/non-HSCs) can differentiate into AT2 and BASC cells that function as progenitor cells. Immediately following bleomycin administration, WT recipient mice underwent intravenous administration of VSELs or non-VSEL/non-HSCs from SPC H2B-GFP mice. GFP+ AT2 and BASC were isolated and tested for progenitor activity using in vitro organoid assays. Results: after 21 days in vivo, we observed differentiation of VSELs but not non-VSEL/non-HSCs into phenotypic AT2 and BASC consistent with previous data in irradiated recipients. Subsequent in vitro organoid assays revealed that VSEL-derived AT2 and BASC maintained physiological potential for differentiation and self-renewal. Conclusion: these findings prove that VSELs produce functional BASC and AT2 cells, and this may open new avenues using VSELs to develop effective cell therapy approaches for patients with lung injury.

## 1. Introduction

Regional epithelial stem/progenitor cell populations maintain and regenerate lung tissue in normal physiological and damage repair states. Alveoli are lined by alveolar type 1 epithelial (AT1) cells, which are responsible for gas exchange, and alveolar type 2 (AT2) cells, the main source of surfactant protein secretion including surfactant protein C (SPC). In addition, AT2 cells serve as the facultative stem cell population for alveoli, as they can divide to produce additional AT2 as well as AT1 cells [1]. Upstream of AT2 cells, bronchioalveolar stem cells (BASCs) at the bronchioalveolar duct junctions serve as progenitors for both airway lining epithelial cells and AT2 cells [2,3,4]. BASC coexpress SPC and secretoglobin family 3A member 2 (Scgb3a2) (also called CC10) [2,4]. In response to damage, BASCs proliferate and give rise to bronchiolar Club cells that express CC10 and SPC+ AT2 cells to promote lung repair [4,5]. Both BASC and AT2 cells play critical roles in lung repair [6,7,8,9]. In cases of chronic injury or acute severe lung injury, these endogenous stem/progenitor populations may not be adequate to repair the lung, leading to loss of pulmonary function. Enhancement of adult autologous stem cells to engraft as functional epithelial cells in the lung could lead to improved treatment strategies for lung repair and regeneration.

Previous studies in which WT donor cells were transplanted into SPC-null recipients [10] demonstrated that SPC expressing AT2 can be derived from nonhematopoietic donor-derived bone marrow (BM) cells. In contrast, hematopoietic stem and progenitor cells (HSPCs) did not give rise to lung epithelial cells. Thus, cells in the nonhematopoietic BM fraction can serve as a source of BM-derived AT2 cells [10].

Evidence accumulated that murine adult tissues contain stem cells with a broad differentiation potential [11,12]. The name “very small embryonic-like stem cells” (VSELs) was assigned to a rare population of small cells that were initially isolated from murine BM [13,14] as well as other organs [15]. Also, small lineage negative (lin^−^) cells that do not express CD45 (CD45¯) and display a primitive morphology (high nuclear/cytoplasm ratio and ground glass appearance of the nuclei) were reported in human umbilical cord blood (UCB) [16,17] and mobilized peripheral blood (mPB) [18]. VSELs injected into mice can contribute to hematopoiesis [19] and pulmonary alveolar epithelium [10,20] in appropriate in vivo models.

In prior studies, we demonstrated that following irradiation and transplantation, VSELs are the main BM derived population giving rise to SPC-positive AT2 cells [10]. However, these studies did not address whether the VSELs engraft as AT2 and BASCs that are capable of forming organoids with mature epithelial cells of the airway and alveoli, nor did they address whether VSELs could engraft as lung epithelial cells following bleomycin-induced injury. AT2 cells form organoids containing AT1 and AT2 cells and BASCs give rise to organoids with AT2 and CC10-expressing airway-lining epithelial cells [21]. Here, we test for engraftment of BM-derived cells as organoid-forming AT2 cells and BASC using a model of bleomycin-induced lung injury.

## 2. Material and Methods

### 2.1. Mice

Experiments were approved by the local Bioethics Commission for animal testing, and all treatments in mice were carried out by persons who had approval from the Medical University of Warsaw. Mice were maintained under constant lighting conditions (12 h light and 12 h of darkness), as well as in controlled humidity (55% ± 10%) and temperature (22 °C ± 2 °C). Wild type (C57BL/6Clzd) mice were purchased from Central Animal Laboratory at Medical University of Warsaw. SPC-H2B-GFP BAC transgenic mice expressing H2B-GFP (Histone 2B-Green Fluorescence Protein) nuclear-localized fusion protein from the SPC promoter [22] were received from Carla Kim (Boston Children’s Hospital). In the lungs of these mice in adulthood, approximately 50–63% of AT2 cells express GFP. [22]

### 2.2. Isolation of Murine VSELs and Non-VSELs/Nonhematopoietic Stem Cells by FACS

VSELs (Lin^−^/Sca1^+^/CD45^−^/Ter119^−^) and non-VSEL/non-HSCs (Lin^−^/Sca1^−^) were isolated by multiparameter, live-cell sorting (MoFlo) as described previously [23,24,25]. Briefly, after red blood cell lysis of BM (BD lysing buffer, BD Biosciences), immunofluorescent staining was performed in Dulbecco’s Modified Eagle Medium (DMEM) containing 2% FBS. The following antimouse antibodies were used: PE anti-CD45R/B220 (Clone: RA3-6B2), PE anti-Gr1 (Ly-6G and Ly-6C) (Clone: RB6-8C5), PE anti-TCRβ (Clone: H57-597), PE anti-γδ T Cell (Clone: GL3), PE anti-CD11b (Clone: M1/70), PE and APC anti-TER-119 (Clone: TER-119), APC-Cy7 anti-CD45 (Clone: 30-F11), Biotin anti-Ly-6A/E (Clone: E13-161.7), PE-Cy5 Streptavidin, PE and PE-Cy5 Rat IgG2a, κ (Clone: R35-95), APC-Cy7 Rat IgG2a, κ (Clone: A95-1). After staining, cells were washed and resuspended in PBS containing 2% FBS for FACS.

### 2.3. Intravenous Transplantation of BM Cells Following Intratracheal Bleomycin Administration

Healthy SPC-H2B-GFP mice (8–10-week-old) were used as donors for non-VSEL/non-HSCs and VSELs isolated from the BM of tibias and femurs. Healthy female WT mice (6–8 week-old) were used as recipients of transplanted cells (as illustrated in Figure 1, Panel A). Mice were divided into groups: (1) Bleo: WT mice (*n* = 9) with bleomycin-induced lung injury with saline infusion only as the lung injury control; (2) VSEL: WT mice (*n* = 16) with bleomycin-induced lung injury followed by VSEL administration; (3) non-VSEL/non-HSC: WT mice (*n* = 9) with bleomycin-induced lung injury followed by non-VSEL/non-HSC administration; (4) NB: WT mice (*n* = 8) without bleomycin-induced lung injury or transplanted cells; (5) SPC-GFP: reference donor SPC-GFP mice. Bleomycin (2.5 mg/kg) suspended in 50 μL of sterile PBS was administered intratracheally (IT) into WT mice. Four hours after bleomycin administration, recipient mice were injected IV (retro-orbital plexus) with saline, 5000 VSELs or 100,000 non-VSEL/non-HSCs. The number of cells administered results from the difference in their frequency in the bone marrow. As reported in the past, the number of cells with the VSEL phenotype is very low and declines with age [26]. Therefore, to reflect this difference in the ratio of VSELs to other cell types in the bone marrow, we administered in our study 5000 VSELs and 100,000 non-VSEL/non-HSC, respectively. Moreover, these cell populations were studied only using a model of lung injury, as we showed previously that lung injury is required for the specification of bone marrow-derived lung epithelial cells [27].

### 2.4. Lung Harvest and Lung Single Cell Suspension

For analysis 21 days post-transplant, mice were anesthetized with ketamine/xylazine, followed by thoracotomy and right ventricular perfusion to remove blood cells from the alveolar space as described previously [27]. Briefly, the left and middle lung lobe was tied off and left lobe was processed for paraffin embedding. The remaining lung was inflated with 3 mL collagenase (LS004212, Worthington biomedical corporation, Lakewood, NJ, USA) in Dulbecco’s modified Eagle’s medium (DMEM) followed by 1% low melting agarose (AB00981, American Bio, Natick, MA, USA). Next, the lung was digested with collagenase IV for 30 min at 37 °C, dissociated using gentleMACS tissue dissociator (Miltenyi Biotec, Bergisch Gladbach, Germany) and incubated with DNase (100 units/mL) for 15 min at 37 °C. Cells were then filtered through 100- and 40-µm cell strainers and washed and processed for flow analysis and cell sorting.

### 2.5. Isolation of Murine Alveolar Type II Cells and Bronchioalveolar Stem Cells by FACS and Flow Cytometry Analysis

AT2 and BASC sorting was performed as previously described [28] (the number of mice from which AT2 and BASC were isolated for analysis: Bleo: *n* = 5; NB: *n* = 6; non-VSEL/non-HSC: *n* = 4; VSEL: *n* = 11). Briefly, single lung cells suspension was stained with the following antimouse antibodies: APC anti-CD31 (Clone: MEC 13.3), APC anti-CD45 (Clone: 30-F11), PE anti-Ly-6A/E (Clone: E13-161.7) and PE-Cy7 anti-CD326 (EP-CAM) (Clone: G8.8). After staining, cells were washed once and resuspended in PBS containing 2% FBS. AT2 (CD31^−^, CD45^−^, Ep-CAM^+^, Sca-1^−^), AT2-GFP+ (CD31^−^/CD45^−^/Ep-CAM^+^/Sca-1^−^/GFP^+^), BASCs (CD31^−^, CD45^−^, Ep-CAM^+^, Sca-1^+^) and BASCs-GFP^+^ (CD31^−^, CD45^−^, Ep-CAM^+^, Sca-1^+^, GFP^+^ were sorted using a FACS Aria (Becton Dickinson, Franklin Lakes, USA) (as illustrated in Figure 2). This staining method was also used for flow cytometry (as illustrated in Figure 5, Panel A−C) using a FACSVerse (Becton Dickinson).

### 2.6. Lung Tissue Paraffin-Embedded Sections

The left lung (as illustrated in Figure 1, Panel B) was infused with 1% low melting agarose (AB00981, American Bio, Natick, MA, USA). After cooling, the lung was fixed with 4% neutral buffered formalin, embedded in paraffin. Sections (3 µm) were stained with hematoxylin and eosin (H&E) and Masson Trichrome, examined using a NIB-100 (Alltion, Wuzhou, Guangxi, China) light microscope, and imaged using an ISH 1000 (TUCSEN, Fuzhou, Fujian, PRC) camera and ISCapture V3.0 software.

### 2.7. Lung Organoid Model System

Organoid cultures based on McQualter et al. [29] and Teisanu et al. [30] revised and modified by Barkauskas et al. [8] were conducted in 24-well Transwells with 0.4 µm pores filter inserts (Corning, Corning, NY, USA) in a 24-well tissue plate containing 410 μL of 3D medium. 3D medium contains Dulbecco’s Modified Eagle’s Medium/F12 (88%, *v*/*v*) supplemented with FBS (10%, *v*/*v*), penicillin/streptomycin (0.5%, *v*/*v*), 1 M HEPES (0.1%, *v*/*v*), and insulin/transferrin/selenium (1%, *v*/*v*). FACS sorted GFP positive AT2 cells (3 × 10^3^) and BASCs (1 × 10^3^) (per single 0.4 µm insert) were cocultured with (1 × 10^5^) MLG cells (Mlg2908, ATCC CCL-206) in Matrigel (Growth Factor Reduced (GFR) Basement Membrane Matrix, Phenol Red-free, *LDEV-free; #356239; Corning, New York, USA) that was prediluted 1:1 with 3D medium. Cultures were incubated at 37 °C in a humidified incubator (5% CO2), with medium replacement every other day. After 21 days, organoids were analyzed with an Olympus FV1000 confocal microscope without any staining. Endogenous GFP signal (presented only by cells that express SPC, which would be AT2 and BASC) was visualized and images were taken with 10× and 20× objectives. Image analysis was performed using ImageJ 1.53e software.

### 2.8. Immunofluorescence Staining on Lung Tissue Sections and Organoids

The organoids were then formalin fixed and paraffin embedded. Sections were deparaffinized with xylene, treated with antigen retrieval solution (1 g NaOH, 2.1 g citric acid in 1 L of H_2_O) for 20 min in steam, cooled to room temperature, permeabilized with PBS-0.2% Triton for 15 min, and blocked with 10% donkey serum in PBS. Sections were stained with monoclonal rabbit antithyroid transcription factor 1 (TTF1, ab76013; Abcam, Cambridge, MA) or rabbit anti-CC10 (ab40873, Abcam, Cambridge, MA, USA) at 1:500 followed by Alexa 594-conjugated donkey antirabbit secondary antibody (ab150076, Abcam, Cambridge, MA, USA) at 1:500, goat anti-GFP (ab5450; Abcam, Cambridge, MA, USA) at 1:500 followed by Alexa 488-conjugated donkey anti-goat secondary antibody (ab150129, Abcam, Cambridge, MA, USA) at 1:500. Images were taken at 10×, 20×, and 40× with an Olympus FV1000 confocal microscope. Image analysis were performed using ImageJ software.

### 2.9. qRT-PCR for GFP Expression

The middle lobe of the right lung (as illustrated in Figure 1, Panel B) was harvested to isolate mRNA. GFP mRNA was quantified by real-time PCR using SYBR Green Super Mix (#1725124, Bio-Rad). Primers were GFP forward: 5′-CCACATGAAGCAGCACGAC-3′, GFP reverse: 5′-CACGAACTCCAGCAGGACCATG-3′. Quantification was calculated using the comparative ΔCT method and normalized to B2-Microglobulin (primer forward: 5′-CATACGCCTGCAGAGTTAAGCA-3′, primer reverse: 5′-GATCACATGTCTCGATCCCAGTAG-3′). All PCRs were performed using the following conditions: predenaturation at 95 °C for 3 min, 40 cycles of denaturation at 95 °C for 10 s, and annealing at 60 °C for 60 s.

### 2.10. Statistical Analysis

Data were analyzed with Prism 6 (GraphPad Software, San Diego, CA, USA) as mean ± SD. Statistical analysis was performed by using two-tailed unpaired Student’s *t*-test for comparison of differences between two groups. *p* < 0.05 was considered statistically significant.

## 3. Results

### 3.1. Transplantation of VSELs Versus Non-VSEL/Non-HSCs Cells into Bleomycin-Induced Lung Injured Mice

We tested different cell populations postbleomycin administration for their ability to differentiate into functional epithelial cells of the lung, specifically AT2 cells and BASC. (as illustrated in Figure 1). To validate bleo-induced pulmonary injury, histopathological assessment was performed on lungs d21 post bleo. Consistent with prior publications [31], there was thickening of the alveolar septa and varying degrees of injury of the airway wall and of the peribronchiolar spaces in association with inflammatory cell infiltration. In severely affected regions, there was replacement of normal lung tissue with collagen.

Representative data showing the flow cytometric isolation of the donor cell populations used is shown in Figure 2. VSELs are Lin^−^Sca^+^CD45^−^ and the nVSEL/nHSC population contains all of the remaining cells (including Lin^+^/CD45^+^) except for HSPCs in the Lin^−^Sca^+^CD45^+^ gate. Recipient WT mice treated with bleomycin received no cells (Bleo only), or intravenous administration of 5000 VSELs or 100,000 non-VSEL/non-HSCs from SPC H2B-GFP mice (as illustrated in Figure 1A). Administration of VSELs or non-VSEL/non-HSCs did not significantly affect Bleo-induced pathophysiology at day 21 (as illustrated in Figure 1C,C’,D’), and later timepoints were not assessed.

### 3.2. Detection of GFP in Lungs of Mice Receiving VSELs Versus HSPCs

To determine whether AT2 cells engraft from H2B-GFP^+^ donor cells delivered intravenously after bleomycin-induced injury, we assessed for GFP expression using quantitative RT-PCR on lung tissue lysate from recipients in all experimental groups (*n* = 3 experiments, Figure 3). The mean GFP mRNA expression in age-matched SPC-GFP mice (same as bone marrow donor animals) was used as the reference point of 100%. In both of the negative control groups (Bleo (*n* = 9) and NB (*n* = 8)), the average GFP mRNA expression signal was 0.13 ± 0.24% and 0.01 ± 0.01% of positive control mice. In contrast, the VSEL group (*n* = 10) had an average GFP mRNA expression 9.57 ± 9.00% of that in the positive control mice (*p* = 0.012 and *p* = 0.009 in comparison to that of NB and Bleo groups, respectively). In the non-VSEL/non-HSCs recipient group (*n* = 9), the GFP mRNA expression was 0.04 ± 0.04% of positive control mice (no statistically significant difference from the negative control Bleo group). Thus, RT-PCR analysis of GFP reveals that WT mice transplanted with VSELs from SPC-H2B-GFP mice have GFP expressing cells in their lungs.

### 3.3. Detection of GFP^+^ Alveolar type II Cells Derived from VSELs by Confocal Microscopy on Tissue Sections

To assess whether H2B-GFP^+^ donor cells exhibit AT2-specific morphology and location within lung tissue, we performed microscopy on lung tissue sections for coexpression of TTF1 and GFP proteins. High-power confocal microscopy images and cross-sections through a series of images taken in different planes (z-stack) of AT2 cells from positive control SPC-GFP mice confirm that TTF1 is colocalized with GFP in the nucleus (as illustrated in Figure 4) of AT2 cells, as reported previously [22], and that these AT2 cells are localized within the alveoli (as illustrated in Figure 4). This staining pattern and associated morphology were used as a standard to identify GFP and TTF1 double positive cells in the lungs of mice from each experimental group sacrificed 21 days after bleomycin treatment.

In negative control mice with bleomycin-induced lung injury but without cell transplant (Bleo group), no GFP^+^ AT2 cells were observed (as illustrated in Figure 4A, bottom row). Similarly, WT mice in the NB group showed no GFP^+^ cells (data not shown). In the VSEL recipients, cells with colocalization of TTF1 and GFP indicating donor derived AT2, were observed (as illustrated in Figure 4A, second row). Of note, VSEL-derived AT2 GFP^+^ cells often occurred in clusters (as illustrated in Figure 4B). In contrast to the VSEL group, no GFP^+^ cells were detected in the lungs of mice in the non-VSEL/non-HSC group. VSEL-derived cells formed discrete clusters of AT2 cells. Because these clusters were distributed unevenly throughout the lung, perhaps due to repair processes, quantification based on thin sections of lungs from mice from individual experimental groups would not be objective. We therefore quantitated donor derived AT2 cells and BASC in the lungs of recipient mice using flow cytometry.

### 3.4. Detection of GFP^+^AT2 Cells and BASC in the Lung Using Flow Cytometry

Using flow cytometry, at day 21, single cell preparations from the postcaval lobes (as illustrated in Figure 1B) were analyzed for GFP^+^ AT2 cells and BASCs (gating strategy, as illustrated in Figure 5A). Representative plots of GFP^+^ AT2 and BASC from each experimental group are shown (as illustrated in Figure 5B). For normalization, the percentage of GFP^+^ AT2 cells and BASCs in positive control SPC-H2B-GFP mice was 34.16 ± 1.75% and 49.62 ± 9.62%, respectively, which we normalized to 100% (as illustrated in Figure 5D). The total number of AT2 cells and BASCs did not differ statistically significantly between groups (data not shown). Note that negative control animals (bleomycin only, *n* = 7) have low background levels of AT2 cells in the GFP gate (the actual percentage was 0.20 ± 0.11%, which was normalized to 0.57 ± 0.31% of that in the positive control mice, Figure 5, Panels B and D). These cells in the GFP^+^ gate are likely due to low level autofluorescence, which was reported for AT2 and BASC [32,33]. To normalize the contribution of SPC-GFP derived epithelial cells, we assessed the percentage of GFP^+^ AT2 and GFP^+^ BASC cells in relation to the total AT2 and BASC per one animal, respectively. Analysis (*n* = 3 experiments) of VSEL recipient mice (*n* = 13) revealed significantly more GFP^+^ AT2 cells (3.08 ± 1.61%) than in the bleo only group (*p* = 0.001). There was no statistically significant difference in GFP^+^ AT2 cells in recipients of non-VSEL/non-HSCs (*n* = 6; 0.55 ± 0.57%) compared to that of the Bleo group (*p* = 0.92). The background level of GFP^+^ cells in the BASC gate from the Bleo control group (*n* = 7) was 1.64 ± 1.13% (as illustrated in Figure 4, Panels C–D). In the VSEL group (*n* = 13), 9.10 ± 6.36% (*p* = 0.009 in comparison to Bleo group) of BASC were GFP^+^, whereas only 1.59 ± 0.969% (no statistically significant difference compared to that of Bleo group) of BASCs were GFP^+^ in lungs of the non-VSEL/non-HSC group (*n* = 6).

### 3.5. Detection of GFP^+^ Epithelial Contribution of Transplanted VSELs Versus Non-VSEL/Non-HSCs in Organoids

The presence of clusters of SPC-GFP^+^ cells in the VSEL recipients suggests that donor derived AT2 cells and/or BASC may serve as functional progenitors. To test directly whether VSEL derived AT2 cells and/or BASC [8,29,30] are capable of proliferation and differentiation, we grew organoids from flow sorted GFP^+^ and GFP^−^ AT2 cells and BASCs from the lungs of mice in each experimental group. Organoids initiated with AT2 (3000) cells or BASC (1000) isolated from SPC-GFP donor mice were used as positive controls. After 21 days of culture, organoids were evaluated without additional staining to assess the endogenous GFP expression from SPC-producing epithelial lung cells, i.e., AT2 and BASCs. A clear GFP signal was detected in organoids derived from both AT2 and BASCs from positive control SPC-GFP mice (as illustrated in Figure 6). Mice from the VSEL group yielded GFP^+^ organoids, whereas no GFP^+^ organoids were derived from the negative control Bleo mice (not shown) or Bleo-treated mice infused with non-VSEL/non-HSCs. Quantification (as illustrated in Figure 6C,D) revealed no significant differences in the number of organoids formed (as illustrated in Figure 6C) from GFP^−^ AT2 cells between the groups. In contrast, the percentage of organoids that were GFP^+^ showed remarkable differences. Among organoids seeded from GFP^+^ AT2 cells from the VSEL group, 96% emitted an endogenous GFP signal (as illustrated in Figure 6D). Also, among organoids seeded from GFP^-^ AT2 cells, approximately 75% from the SPC-GFP mice and 22% from the VSEL group grew GFP^+^ organoids reflecting activation of the transgene in cells derived from SPC-GFP control and donor mice during culture, respectively. In contrast, in the non-VSEL/non-HSC and Bleo groups, no GFP^+^ organoids were observed.

Similar to the organoids formed from AT2 cells, among organoids plated with GFP positive BASCs, 94% were GFP-positive in the VSELs group (as illustrated in Figure 6D). Among organoids plated from GFP-negative BASCs from the VSEL group, 25% were GFP-positive. Organoids from the Bleo-only and non-VSEL/non-HSC groups did not show a GFP signal.

To confirm that GFP^+^ cells in the organoids were AT2 cells, immunofluorescence for GFP and TTF1 was performed. Microscopy clearly showed colocalization of TTF1 with GFP in SPC-H2B-GFP controls (not shown) and in the VSEL recipient group (as illustrated in Figure 7A). Such colocalization was not observed for either the non-VSEL/non-HSCs or Bleo only groups. This indicates that VSEL-derived AT2 cells have the same physiological capabilities for proliferation and differentiation as endogenous AT2 cells. To confirm that the phenotypic BASC identified in the VSEL recipients were functional, organoids derived from FACSorted BASC were costained for airway epithelial-like cells with anti-CC10 and AT2 cells with anti-GFP. Figure 7 (Panel B) shows the presence of CC10^+^ and GFP^+^ cells within the organoids consistent with the initiating cells for these organoids being functional BASC. Some single cells stained for both CC10^+^ and GFP^+^ cells, indicating that BASC cells of VSEL origin can self-renew, as well as differentiate into other types (e.g., airway epithelial cells) of lung cells.

## 4. Discussion

The aim of our study was to determine whether bone-marrow-derived VSELs transplanted into mice after lung injury can differentiate into lung epithelial cells that can serve as functional progenitors. VSELs can differentiate into lung epithelial cells post-transplantation following irradiation [10]. Here, we demonstrate the engraftment of VSEL-derived lung epithelial progenitors in a mouse model with bleomycin-induced lung injury. Kassmer et al. [10] showed that postirradiation transplanted VSELs engraft approximately 4% of AT2 cells. We revealed that VSELs transplanted into mice with bleomycin-induced lung damage may engraft to higher levels. The higher degree of engraftment observed may be due to the use of more VSELs for transplantation (we transplanted 5000 VSELs, and Kassmer et al. used 900–1500 VSELs), or a different form of lung damage (we used bleomycin [34], and Kassmer et al. used radiation).

Using immunofluorescence, flow cytometry, and functional organoid assays, we showed that in the bleomycin-induced lung injury model, VSELs differentiate into both AT2 and BASC. The presence of a low number of GFP-positive cells in the negative control Bleo group may be due to autofluorescence of some AT2 cells [4]. In previous studies using a mouse model of bleomycin-induced lung injury, BASCs differentiated from mesenchymal cells isolated from bone marrow [5].

The presence of clusters of donor-derived GFP-positive AT2 cells by confocal microscopy is consistent with the observations of Desai et al. [35], who described the mechanism of the formation of foci of immature AT2 cells. They found that foci were similar in size and progressively enlarged, indicating derivation from a single ‘founder’ AT2 cell.

After confirming the presence of AT2 cells and BASCs of VSEL origin, the next step was to assess their in vitro proliferative and differentiation potential. We therefore established organoid cultures from isolated GFP-positive AT2 cells and BASCs. This technique is widely used and described in the literature as a method of assessing the physiological ability of individual lung progenitor cells to proliferate and differentiate lung epithelial cells [36,37]. We observed GFP^+^ cells in both AT2 and BASC cell-derived organoids.

In our studies, we showed for the first time that not only VSELs differentiate in the mouse model of bleomycin-induced lung injury in alveolar type 2 cells and bronchioalveolar stem cells, but also that the pulmonary epithelial cells proliferated from VSELs fully retain their physiological proliferative potential. This gives grounds for further research to develop an effective method of cell therapy in patients with lung damage. The current experiments were not designed to assess therapeutic effects of VSELs; however, future studies are warranted to address this.

We are aware that VSELs are still considered controversial. We believe that other groups should embrace studies of this cell type. Although we are aware of high-profile paper questioning existence and primitive character of VSELs in the bone marrow [38], we strongly encourage investigators to contact us, if necessary, for assistance in VSEL isolation/purification using the sorting protocol described in detail [25]. At the same time, it is important to emphasize that many independent investigators not only confirmed the existence of VSELs, but also their high proliferative and differentiation potential [17,19,39,40,41,42,43,44,45].

## Figures and Tables

**Figure 1 cells-10-01570-f001:**
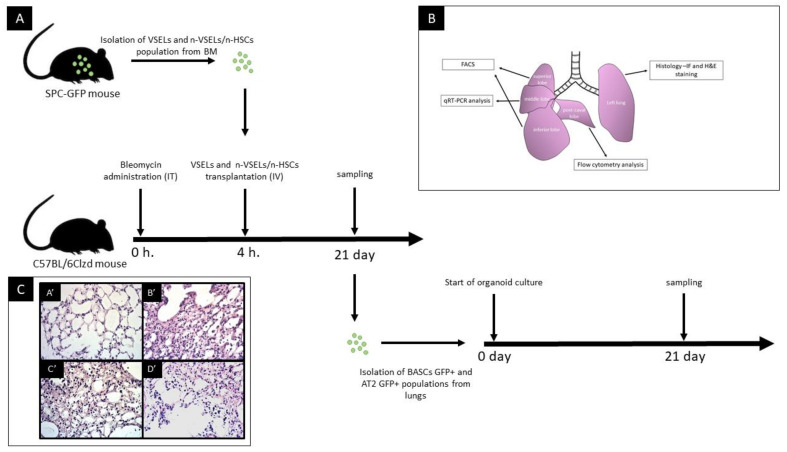
Panel **A**: schematic of study design. IT—intratracheal route of administration; IV—intravenous route of administration. Panel **B**: schematic of tissue harvested for different analyses as indicated. Panel **C**: representative images of lung tissue sections from 21 days after bleomycin administration, paraffin embedded and H&E stained. Images were taken at 100× magnification. Histological picture of lung biopsies varies over time from the acute phase followed by fibrotic changes—mainly bronchiolar and peribronchiolar. In our histopathological study assessment performed at d21, time of chronic phase injury characterized by thickening of alveolar septa, varying degrees of thickening of airway wall and peribronchiolar spaces in association with inflammatory cell infiltration. **A’**: NB group (untreated control). Normal histological appearance of lung.; **B’**: Bleo only mouse; **C’**: Bleo + VSEL mouse; **D’**: Bleo + non-VSEL/non-HSC mouse.

**Figure 2 cells-10-01570-f002:**
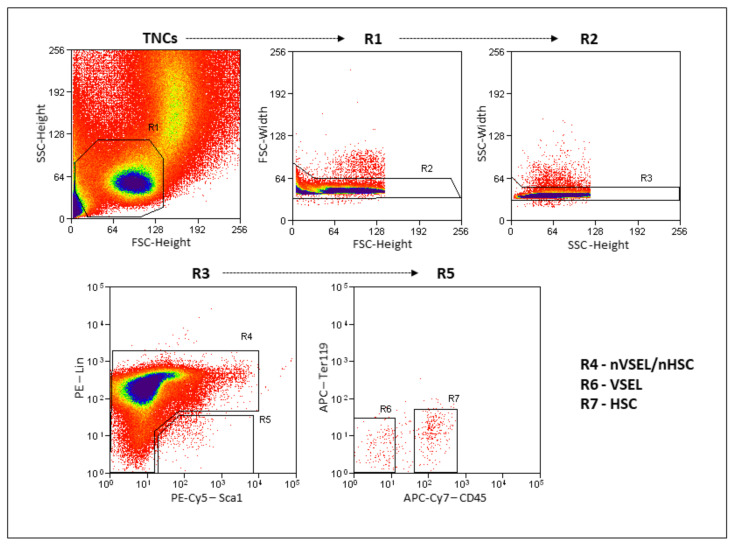
Gating strategy for VSEL and non-VSEL/non-HSC isolation from murine bone marrow. Total nucleated cells (TNCs) are visualized on FSC vs. SSC dot-plot showing. Gates R2 and R3 remove doublets/clusters. Single-cell fraction from gate R3 is further visualized on Sca-1 vs Lin dot plot where gate R4 represents non-VSEL/non-HSC cell population and gate R5 is Sca-1^+^/Lin^−^. Gated region R5 further separated in CD45 vs Ter119 dot plot, separating nonhematopoietic CD45^−^ fraction (VSELs—R6) and hematopoietic CD45^+^ fraction (HSCs—R7).

**Figure 3 cells-10-01570-f003:**
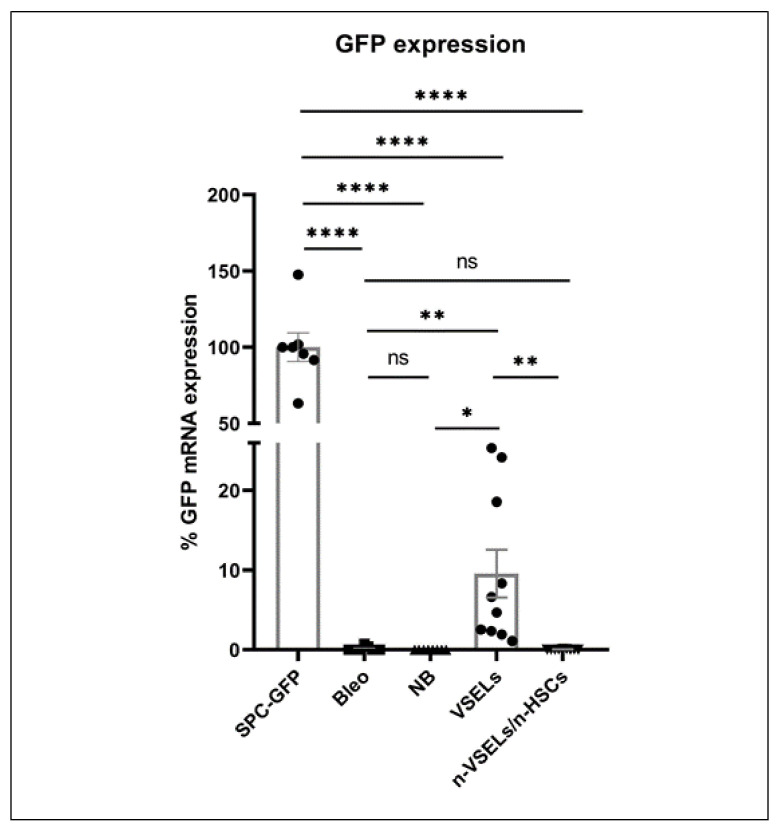
Relative GFP mRNA expression. Lung tissue from SPC-GFP (donor) mice was used as the reference point for GFP expression. Y axis shows relative ΔCT. Statistically significant differences between Bleo group and selected populations are indicated. Results are from 3 experiments. Mean and SD indicated. * *p* <0.05, ** *p* < 0.01), **** *p* < 0.0001.

**Figure 4 cells-10-01570-f004:**
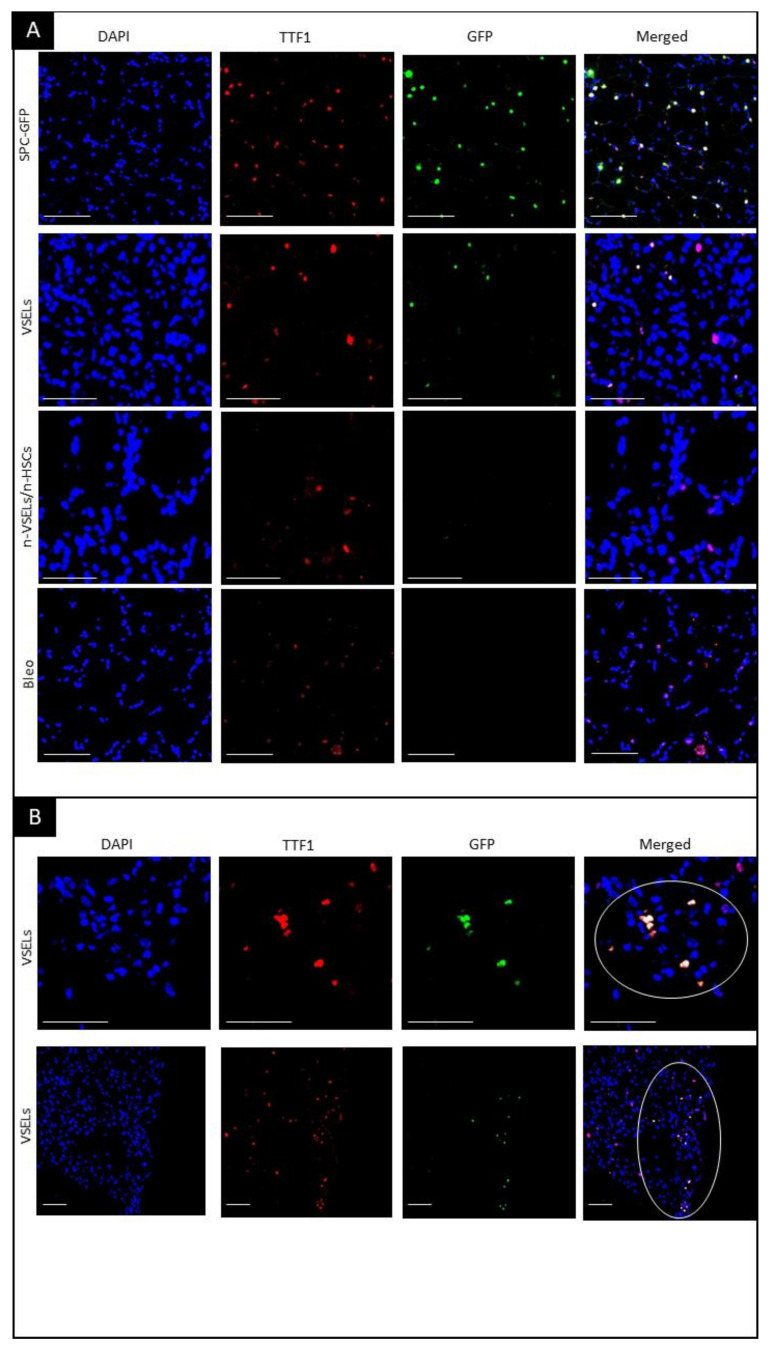
Visualization of GFP^+^/TTF1^+^ AT2 cells in transplant recipients. Fluorescence microscopy on paraffin embedded lung tissue sections. Blue—DAPI, Green—GFP, Red—TTF1, and merged colors. Scale bars represent 100 µm. Panel **A**: Top row: colocalization of DAPI, GFP and TTF1 indicates AT2 cells. Second row: GFP^+^ AT2 cells derived from GFP-positive donor VSELs cells. Third and fourth rows: no GFP^+^ cells. Olympus FluoView V1000 confocal microscope, 40× original magnification (top and bottom row) and 42× digital zoom (two middle rows). Panel **B**: representative images of clusters of GFP^+^/TTF1^+^ VSEL-derived AT2 cells. GFP^+^ AT2 cell clusters in lung tissue from Bleo^+^/VSELs^+^ recipients are marked with a circle. Olympus FluoView V1000 confocal microscope, 45× and 20× original magnification (upper and lower rows, respectively).

**Figure 5 cells-10-01570-f005:**
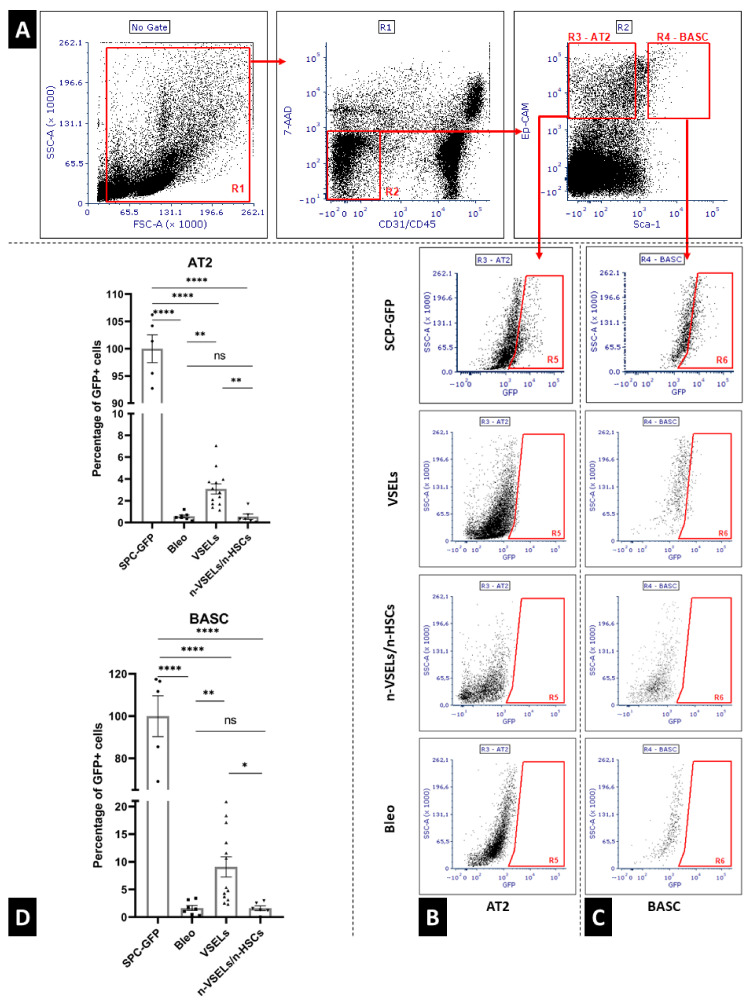
Panel **A**: gating strategy for flow cytometry analysis of viable (7AAD negative) Alveolar Type 2 cells (CD31^−^/CD45^−^/Ep-CAM^+^/Sca-1^−^) and bronchoalveolar stem cells (CD31^−^/CD45^−^, Ep-CAM^+^/Sca-1^+^); Panel **B**: representative GFP plots for Alveolar Type 2 cells from each mouse group; Panel **C**: representative GFP plots for bronchioalveolar stem cells from each mouse group. Panel **D**: quantification of GFP^+^ AT2 (top) and BASC (bottom). Values are standardized to mean GFP positive cells detected in respective population of SPC-GFP donor mice (*n* = 5), treated as 100%. Significant differences between experimental groups are indicated with asterisks. Results are from three experiments. Mean and SD indicated by black lines. * *p* < 0.05, ** *p* < 0.01, **** *p* < 0.0001.

**Figure 6 cells-10-01570-f006:**
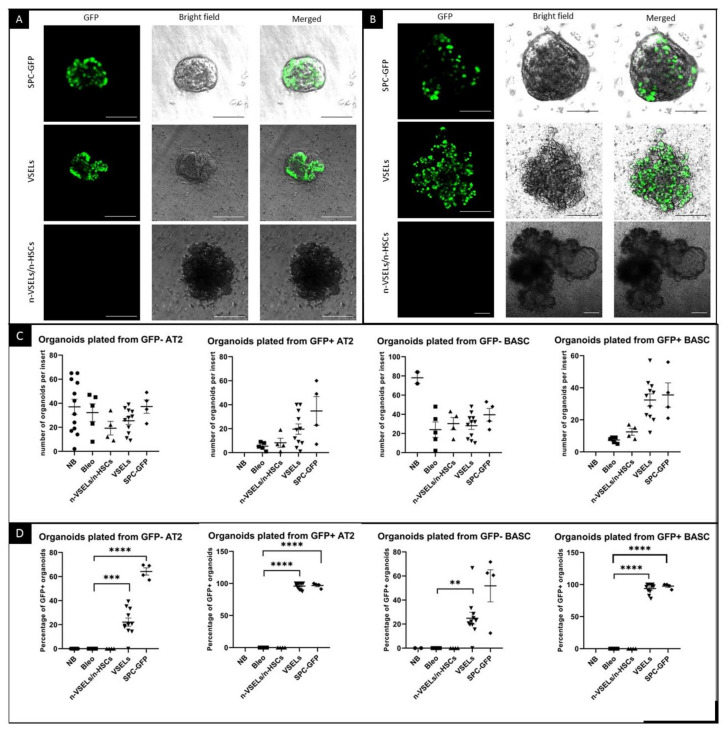
Endogenous GFP fluorescence in organoids and quantification of cultured organoids. Photographs of unstained organoids established from sorted AT2 cells (**A**) and BASCs (**B**) isolated from SPC-GFP donor mice as well as VSEL and non-VSEL/non-HSC recipients as indicated. Photographed organoids are in tissue culture inserts, and green signal is from endogenous GFP. Each image shows an overlay of brightfield and GFP images from the same field of view. Scale bars represent 100 µm in Panel **A** and two upper rows of Panel **B**. Scale bars in bottom row of Panel B represent 200 µm. Quantification was done before organoids were removed from inserts for sectioning. Panel **C.** Graphs showing number of organoids grown for each mouse. Panel **D.** Graphs show percentage of organoids with endogenous GFP signal for each mouse. ** *p* < 0.01, *** *p* < 0.05 **** *p* < 0.0001.

**Figure 7 cells-10-01570-f007:**
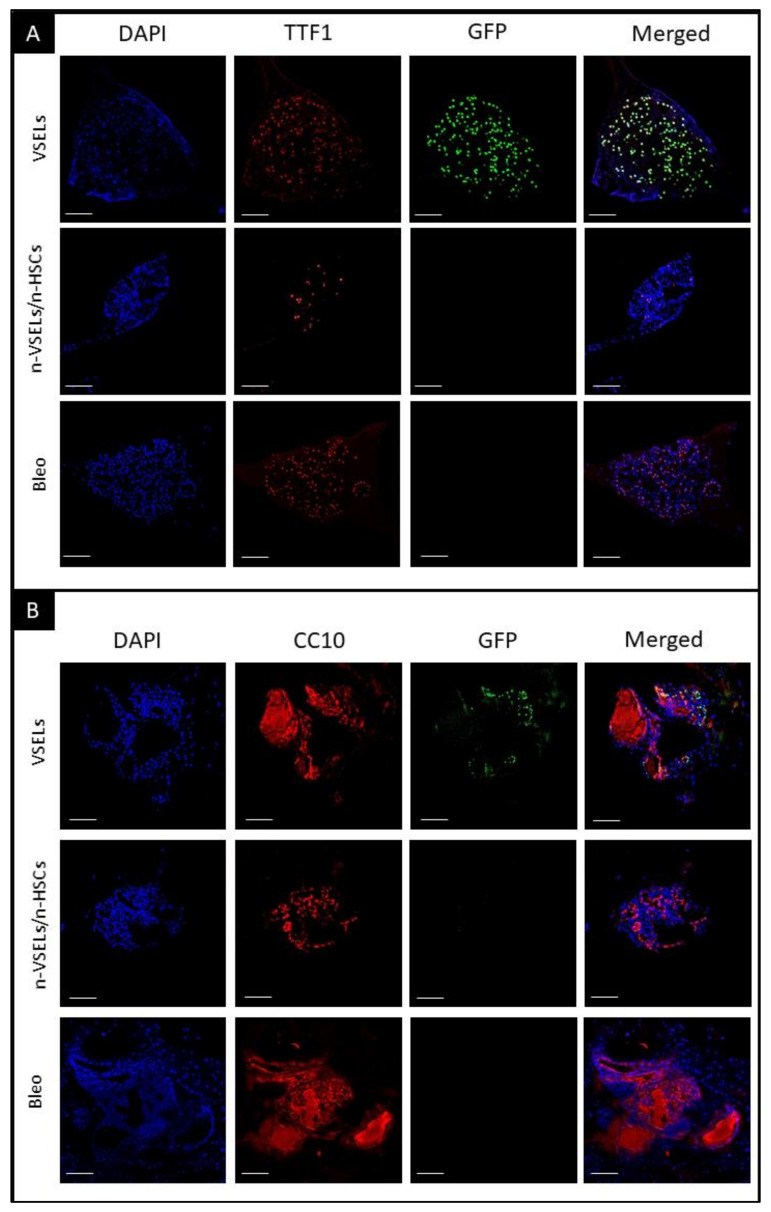
Detection of airway epithelia and AT2 cells in organoids. Immunofluorescent staining of representative organoid sections from each of transplant groups as indicated. Scale bars represent 100 µm. Panel **A**: blue—DAPI, red—TTF1, green—GFP, and merged. Colocalization of DAPI, GFP, and TTF1 indicates donor VSEL derived AT2 cells, representing maintenance of proliferation and differentiation; Panel **B**: blue—DAPI, red—CC10, green—GFP, and merged colors. Colocalization of DAPI, GFP and CC10 indicates that donor VSEL-derived BASCs self-renew and form organoids, which proves their full physiological abilities. Note that CC10^+^ cells are GFP^−^, as GFP is expressed from the SPC promoter. CC10 is cytoplasmic and secretory protein what can be observed in stained Matrigel (cell-free, no DAPI nuclei) in addition to individual cells.

## Data Availability

Raw data is stored on local disks in the Department of Regenerative Medicine and is available on request. In this matter, please contact the corresponding author.

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
