# Peer review of "Bone Marrow-Derived VSELs Engraft as Lung Epithelial Progenitor Cells after Bleomycin-Induced Lung Injury"

_cells, 2021, doi:10.3390/cells10071570_

Round 1
Reviewer 1 Report
The paper is well written and designed. The presented experiments were performed in a rigorous manner.
To better comprehend the full work authors should insert a graphical abstract or a picture with the study design that resume the entire paper.
Author Response
We would like to thank the reviewer for such a favorable opinion. We agree that a graphic abstract will enhance the manuscript, and have added a graphic abstract.
Reviewer 2 Report
Ciechanowicz et al. continue the story of the VSELs, report their engraftment and trans-differentiation into epithelial progenitors in a bleomycin lung injury model. The manuscript is modestly incremental; for the most part it confirms many data previously put forth by some of the co-authors, which is also of merit, would be of greater value yet, though, if the confirmation came from outside the group. Clearly, VSEL cells contribute to lung tissue three weeks after administration. The GFP mRNA signal strength of almost 10% of H2B-GFP+ mice indicates markedly higher contribution than can be explained by VSEL trapping and persistence alone, i.e. the cells seem to have proliferated in situ. And indeed cells with a AT2-like phenotype expressing GFP are identified, thus VSEL seem to have differentiated into mature lung epithelial cells. Despite their quantitative relevant contribution, an effect on lung remodeling is not apparent, in that histology of Bleomycin+VSEL lungs shown similar disruption of normal lung tissue as the other Bleomycin-treated groups.
Fig. 2, addition of a concurrent nuclear dye control (a small aliquot of the flow cytometry sample – I appreciate that cells stained with nuclear dye would not be useful for downstream experiments) would increase confidence that the events are diploid cells. Of course the transplant data shown here for the Bleomycin lung injury model strongly suggest that they are cells.
Two related issues should find their way into the discussion: The concept of the VSEL has received modest attention from outside the Kentucky group, and as a result the manuscript is extremely self-referential. The authors should pro-actively acknowledge both.
Author Response
Ad. Fig. 2, addition of a concurrent nuclear dye control (a small aliquot of the flow cytometry sample – I appreciate that cells stained with nuclear dye would not be useful for downstream experiments) would increase confidence that the events are diploid cells. Of course the transplant data shown here for the Bleomycin lung injury model strongly suggest that they are cells.
We would like to thank the Reviewer for this remark. Cell viability was checked optically using microscopy. Only cells with a viability of at least 95% were used for the experiment. We agree with the Reviewer that the model of experiment used and the possible content of dead cells among the transplanted cells could affect the result of the experiment.
Ad. Two related issues should find their way into the discussion: The concept of the VSEL has received modest attention from outside the Kentucky group, and as a result the manuscript is extremely self-referential. The authors should pro-actively acknowledge both.
Thank you for this comment. We agree with the Reviewer that the concept of VSELs is not yet well published by other groups. We have added the following to the discussion:
“We are aware that VSELs are still considered highly controversial. We think that other groups should embrace studies of this cell type. We are aware of high profile papers suggesting that VSELs do not exist, and encourage investigators to contact us for assistance in VSEL isolation for independent studies on this important cell population. At the same time, it is important to note that many independent research centers around the world not only confirmed the existence of VSELs, but also their high proliferative potential.”
We would also like to point out that every year, a few more independent and non-affiliated research centers publish scientific articles about them. Below are just a few selected sample papers:
- Isolation of a novel embryonic stem cell cord blood-derived population with in vitro hematopoietic capacity in the presence of Wharton's jelly-derived mesenchymal stromal cells. Gounari E, Daniilidis A, Tsagias N, Michopoulou A, Kouzi K, Koliakos G.
2019 Feb;21(2):246-259. doi: 10.1016/j.jcyt.2018.11.006. Epub 2018 Dec 4.
PMID: 30522805 - VSELs Maintain their Pluripotency and Competence to Differentiate after Enhanced Ex Vivo Expansion. Lahlil R, Scrofani M, Barbet R, Tancredi C, Aries A, Hénon P.
Stem Cell Rev Rep. 2018 Aug;14(4):510-524. doi: 10.1007/s12015-018-9821-1. - A Novel Method for Isolation of Pluripotent Stem Cells from Human Umbilical Cord Blood.
Monti M, Imberti B, Bianchi N, Pezzotta A, Morigi M, Del Fante C, Redi CA, Perotti C.
Stem Cells Dev. 2017 Sep 1;26(17):1258-1269. doi: 10.1089/scd.2017.0012. Epub 2017 Jun 4. PMID: 28583028 - HIF-2α and Oct4 have synergistic effects on survival and myocardial repair of very small embryonic-like mesenchymal stem cells in infarcted hearts. Zhang S, Zhao L, Wang J, Chen N, Yan J, Pan X. Cell Death Dis. 2017 Jan 12;8(1):e2548. doi: 10.1038/cddis.2016.480.
- Migration of Bone Marrow-Derived Very Small Embryonic-Like Stem Cells toward An Injured Spinal Cord. Golipoor Z, Mehraein F, Zafari F, Alizadeh A, Ababzadeh S, Baazm M.
Cell J. 2016 Winter;17(4):639-47. doi: 10.22074/cellj.2016.3836. Epub 2016 Jan 17. - Molecular and phenotypic characterization of CD133 and SSEA4 enriched very small embryonic-like stem cells in human cord blood. Shaikh A, Nagvenkar P, Pethe P, Hinduja I, Bhartiya D. 2015 Sep;29(9):1909-17. doi: 10.1038/leu.2015.100. Epub 2015 Apr 17.
- Bone-marrow-derived very small embryonic-like stem cells in patients with critical leg ischaemia: evidence of vasculogenic potential. Guerin CL, Loyer X, Vilar J, Cras A, Mirault T, Gaussem P, Silvestre JS, Smadja DM. Thromb Haemost. 2015 May;113(5):1084-94. doi: 10.1160/TH14-09-0748. Epub 2015 Jan 22.
- Hepatic regenerative potential of mouse bone marrow very small embryonic-like stem cells.
Chen ZH, Lv X, Dai H, Liu C, Lou D, Chen R, Zou GM. J Cell Physiol. 2015 Aug;230(8):1852-61. doi: 10.1002/jcp.24913. - Identification of a distinct small cell population from human bone marrow reveals its multipotency in vivo and in vitro. Wang J, Guo X, Lui M, Chu PJ, Yoo J, Chang M, Yen Y. PLoS One. 2014 Jan 17;9(1):e85112. doi: 10.1371/journal.pone.0085112. eCollection 2014
Reviewer 3 Report
Title: Bone marrow-derived VSELs engraft as lung epithelial progenitor cells after bleomycin-induced lung injury – Cells
The authors provide a well-written and well-designed study, in which they show that injected VSEL can home to the injured lung and contribute to the AT2/BASC population, potentially leading to lung repair. These VSEL-derived cells were able to form organoids while retaining their self-renewal and differentiation capacity. The figures are very clear an explanatory.
Minor points:
Page 7, M&M, section mice: The sentence “In these mice, 50-63% of AT2 cells expressing GFP in adults.” is not correct. Please rephrase.
Page 3, section Intravenous transplantation of BM cells following intratracheal bleomycin administration
The following groups were investigated in this study.
1) Bleo: WT mice (n=9) with bleomycin-induced lung injury with saline infusion only as the lung injury control;
2) VSEL: WT mice (n=16) with bleomycin-induced lung injury followed by VSEL administration;
3) n-VSEL/n-HSC: WT mice (n=9) with bleomycin-induced lung injury followed by non-VSEL/non-HSC administration;
4) NB: WT mice (n=8) without bleomycin-induced lung injury or transplanted cells;
5) SPC-GFP: reference donor SPC-GFP mice.
Can the authors explain why a VSEL or n-VSEL/n-HSC were not included? Do the VSEL home to the alveolar areas also without injury (in this case bleomycin)?
Page 6: Authors state that 5,000 VSELs or 100,000 n-VSEL/n-HSCs were administered. Can the authors indicate why these numbers were chosen?
Page 9: Authors state that the GFP signal in the VSEL-treated Bleo-group was uneven. Can the authors further elaborate on this? Was the signal found back in specific regions? From Figure 4, it is unclear whether only alveolar regions were involved, although it is stated in the text. Please include the (matching) H&E stainings in Figure 4.
Major points:
Figure 1: Authors state that representative H&E and Masson Trichrome staining are shown. It is unclear which staining is shown here, and it seems that only H&E is shown. Please add a description of the shown staining and/or include pictures from both stainings.
Further, on page 6, Results, section “Transplantation of VSELs versus non-VSEL/non-HSCs cells into bleomycin-induced lung injured mice” the authors state that there is fibrosis and inflammation visible. A quantification would improve the understanding of the effect of VSELs and non-VSEL/non-HSC treatment on both fibrosis and inflammation: please include.
Figure 2/5:
- The authors define AT2 cells as CD31-/CD45-/Ep-CAM+Sca-1 negative- and BASC as CD31-/CD45-/Ep-CAM+Sca-1 negative+. Can other populations (bronchial, AT1, other) excluded, as these cells may differentiate in other cell types?
- The authors rightfully focus on the AT2 and BASC populations, and show clear presence of GFP+ populations in the VSELs-treated bleo-group. Are any of the other populations (CD45/ CD31/ other) different between the groups? Please include whether these populations are similar or different upon bleo-exposure or treatment.
Figure 6/7:
- Although authors do not characterize the formed organoids, there are differences in organoid morphology (Fig. 6) and non-VSEL/non-HSCs treated mice seem negative for TTF1 (Fig. 7a). Were absolute numbers of AT2/BASC cells different between the various groups (only GFP+ cells are shown in Fig. 5)? Please discuss shortly.
- Please discuss potential differentiation into bronchial organoids.
Discussion:
The authors do not discuss the potential treatment with VSEL. Authors state that levels of fibrosis/inflammation were similar, and do not discuss why not treatment effect was observed. Please discuss this aspect.
Author Response
Page 7, M&M, section mice: The sentence “In these mice, 50-63% of AT2 cells expressing GFP in adults.” is not correct. Please rephrase.
We have corrected the grammar and clarified the meaning of this sentence in the text to "In the lungs of these mice in adulthood, approximately 50-63% of AT2 cells express GFP."
Page 3, section Intravenous transplantation of BM cells following intratracheal bleomycin administration
The following groups were investigated in this study.
1) Bleo: WT mice (n=9) with bleomycin-induced lung injury with saline infusion only as the lung injury control;
2) VSEL: WT mice (n=16) with bleomycin-induced lung injury followed by VSEL administration;
3) n-VSEL/n-HSC: WT mice (n=9) with bleomycin-induced lung injury followed by non-VSEL/non-HSC administration;
4) NB: WT mice (n=8) without bleomycin-induced lung injury or transplanted cells;
5) SPC-GFP: reference donor SPC-GFP mice.
Can the authors explain why a VSEL or n-VSEL/n-HSC were not included? Do the VSEL home to the alveolar areas also without injury (in this case bleomycin)?
We thank the Reviewer for this question. As shown in the earlier work (Herzog et al., Stem Cells 2006), no bone marrow-derived lung epithelial cells are formed in the absence of tissue injury. We have modified the manuscript to include this information in the methods section as follows “Moreover, these cell populations were studied only using a model of lung injury, as we have shown previously that lung injury is required for the specification of bone mar-row-derived lung epithelial cells [27].”
Page 6: Authors state that 5,000 VSELs or 100,000 n-VSEL/n-HSCs were administered. Can the authors indicate why these numbers were chosen?
Thank you for this comment. The difference in the number of VSEL cells (5,000 cells) and n-VSEL/n-HSC (100,000 cells) administered is due to their ratios in the bone marrow. Since VSELs cells are a rare population, the number of n-VSEL/n-HSC cells was matched to their number in the marrow.
In the section "Intravenous transplantation of BM cells following intratracheal bleomycin administration," we have added an explanation to clarify this point to the reader, as follows “The number of cells administered results from the difference in their frequency in the bone marrow. As reported in the past, the number of cells with the VSEL phenotype is very low and declines with age [26]. Therefore, to reflect this difference in the ratio of VSELs to other cell types in the bone marrow, we have administered in our study 5,000 VSELs and 100,000 n-VSEL / n-HSC 100,000, respectively.”
Ad. Page 9: Authors state that the GFP signal in the VSEL-treated Bleo-group was uneven. Can the authors further elaborate on this? Was the signal found back in specific regions? From Figure 4, it is unclear whether only alveolar regions were involved, although it is stated in the text. Please include the (matching) H&E stainings in Figure 4.
Thank you for pointing out this lack of clarity. All of the donor derived AT2 cells were found in the alveolar regions. We meant to indicate that the AT2 cells that proliferated from VSELs were found in lung tissue in clusters – there were regions with none, and regions with many. This is likely due to the nature of the bleomycin-induced injury, which causes heterogeneous lesions/injury to the lung tissue. We have clarified this in the text as follows “VSEL-derived cells formed discrete clusters of AT2 cells. Because these clusters were distributed unevenly throughout the lung, perhaps due to repair processes, quantification based on thin sections of lungs from mice from individual experimental groups would not be objective.”
Figure 1: Authors state that representative H&E and Masson Trichrome staining are shown. It is unclear which staining is shown here, and it seems that only H&E is shown. Please add a description of the shown staining and/or include pictures from both stainings.
Thank you very much for this remark. Figure 1C shows illustrative pictures of H&E stained lung fragments. Masson Trichome staining was also performed on some sections to confirm fibrosis but was not shown since it was only used internally to establish the occurrence of lung lesions.
We have changed the description of Figure 1C, specifying only H&E staining and adding a description of the changes observed as follows “The histological picture of lung biopsies varies over time from the acute phase followed by fibrotic changes - mainly bronchiolar and peribronchiolar. In our histopathological study assessment performed at d21, the time of chronic phase injury characterized by thickening of the alveolar septa, varying degrees of thickening of the airway wall and the peribronchiolar spaces in association with inflammatory cell infiltration.”
On page 6, Results, section “Transplantation of VSELs versus non-VSEL/non-HSCs cells into bleomycin-induced lung injured mice” the authors state that there is fibrosis and inflammation visible. A quantification would improve the understanding of the effect of VSELs and non-VSEL/non-HSC treatment on both fibrosis and inflammation: please include.
We agree with the Reviewer that when examining the impact of VSELs and non-VSEL/non-HSC grafts on repair processes by remodeling injured lung tissue, it would be important to perform pathological quantification of lung tissue injury. However, in view of the purpose of our study, it was not necessary to evaluate the degree of lung injury. Figure 1C shows the H&E staining of lung tissue to prove the occurrence of bleomycin-induced tissue injury.
Figure 2/5:
The authors define AT2 cells as CD31-/CD45-/Ep-CAM+Sca-1 negative- and BASC as CD31-/CD45-/Ep-CAM+Sca-1 negative+. Can other populations (bronchial, AT1, other) excluded, as these cells may differentiate in other cell types?
Thank you for this interesting question. The gating strategy presented in Figure 5 assumes the evaluation of only these two populations (AT2 and BASC), as the mouse model we used allowed the determination of the VSEL origin of these two populations because they express surfactant protein C. The evaluation of other populations, such as Club cells, Alveolar Type 1 cells and others, is a very interesting concept, but the SPC-H2B-GFP mice used by us in our experimental design allow the differentiation of AT2 and BASC cells proliferated from donor cells from endogenous AT2 and BASC cell populations of the transplant recipient.
Figure 2/5:
The authors rightfully focus on the AT2 and BASC populations, and show clear presence of GFP+ populations in the VSELs-treated bleo-group. Are any of the other populations (CD45/ CD31/ other) different between the groups? Please include whether these populations are similar or different upon bleo-exposure or treatment.
As the Reviewer notes, we focused on the AT2 and BASC cell populations. We did not assess other populations because there is no expression of the Surfactant Protein C in other populations of lung cells, and thus GFP is not expressed. No attempt was made to determine the percentages of other cell types comprising the lungs in mice that did or did not receive donor-derived VSEL or non-HSC/non-VSEL.
Figure 6/7:
Although authors do not characterize the formed organoids, there are differences in organoid morphology (Fig. 6)
We agree with this observation that the morphology of organoids is different from one another. We showed this phenomenon earlier in Ciechanowicz A.: Stem Cells in Lungs (Adv Exp Med Biol. 2019;1201:261-274. doi: 10.1007/978-3-030-31206-0_13.). Organoids have irregular shapes depending on their percentage of cellular composition, including self-renewing AT2 and AT1 cells proliferated from them. In the case of organoids grown from BASC cells, additionally, BASC self-renewing and AT2, AT1 and Club cells proliferated from them.
Figure 6/7:
non-VSEL/non-HSCs treated mice seem negative for TTF1 (Fig. 7a).
We would like to thank the Reviewer for indicating this concern. TTF1 signal was observed in the organoids grown from AT2 in the non-VSEL/non-HSC group. We have improved Figure 7 now and visualized it more by slightly increasing the color saturation of this image.
Figure 6/7:
Were absolute numbers of AT2/BASC cells different between the various groups (only GFP+ cells are shown in Fig. 5)? Please discuss shortly.
The total number of AT2 and BASC did not differ statistically significantly between groups (data not shown). But not the total percentages of AT2 and BASC varied between individual mice, perhaps due to normal variability or to the different sizes of the mice. Therefore, to normalize the measurements, we assessed the percentage of GFP+ AT2 and GFP+ BASC cells in relation to the total amounts of AT2 and BASC within each mouse, respectively.
As suggested by the Reviewer, we have included this information in the "Detection of GFP + AT2 cells and BASC in the lung using flow cytometry" section of the results description.
Figure 6/7:
Please discuss potential differentiation into bronchial organoids.
VSELs as pluripotent stem cells may also have the potential to differentiate into bronchial stem cells that would give rise to bronchial organoids. Nevertheless, due to the applied experimental design and the unique SPC-H2B-SPC mouse model, we analyzed by isolating only positive GFP cells for culture initiation, i.e., those also expressing SPC, i.e., a protein characteristic only for AT2 cells and BASCs.
Thanks to this solution, we were sure that the organoids analyzed were cultures initiated by AT2 cells and BASCs, respectively.
Discussion:
The authors do not discuss the potential treatment with VSEL. Authors state that levels of fibrosis/inflammation were similar, and do not discuss why not treatment effect was observed. Please discuss this aspect.
We agree with the Reviewer that we have not discussed the potential use of VSELs in cell therapies in the manuscript. In the presented manuscript, we focused on showing that the transplanted VSELs not only settle in injured lungs and differentiate into lung progenitor cells (AT2 and BASCs), but also progenitors proliferated from VSEL cells are fully functional.
The design of the used experiment assumed the demonstration of this aim. We used only very small numbers of VSELs, and selected a time of assessment of just 21 days from VSELs cell transplant to harvest. In future studies, we plan on using different doses of VSELs administered at different times after different degrees of injury in order to assess for potential statistically significant increases in tissue repair.
We also added to the discussion, "The current experiments were not designed to assess therapeutic effects of VSELs, however future studies are warranted to address this.”
Round 2
Reviewer 3 Report
Thanks to the authors for addressing all raised issues and good luck with the follow-up experiments.